# Resin-Based Bulk-Fill Composites: Tried and Tested, New Trends, and Evaluation Compared to Human Dentin

**DOI:** 10.3390/ma15228095

**Published:** 2022-11-15

**Authors:** Nicoleta Ilie

**Affiliations:** Department of Conservative Dentistry and Periodontology, University Hospital, Ludwig Maximilian University of Munich, Goethestr. 70, 80336 Munich, Germany; nilie@dent.med.uni-muenchen.de

**Keywords:** resin-based composites, bulk-fill, Vickers and Martens hardness, indentation modulus

## Abstract

A more-and-more-accepted alternative to the time-consuming and technique-sensitive, classic, incremental-layering technique of resin-based composites (RBCs) is their placement in large increments. The so-called bulk-fill RBCs had to be modified for a higher polymerization depth and already have a 20-year history behind them. From the initial simple mechanisms of increasing the depth of cure by increasing their translucency, bulk-fill RBCs have evolved into complex materials with novel polymerization mechanisms and bioactive properties. However, since the materials are intended to replace the tooth structure, they must be comparable in mechanical behavior to the substance they replace. The study compares already established bulk-fill RBCs with newer, less-studied materials and establishes their relationship to dentin with regard to basic material properties such as hardness and indentation modulus. Instrumented indentation testing enables a direct comparison of tooth and material substrates and provides clinically relevant information. The results underline the strong dependence of the measured properties on the amount of filler in contrast to the small influence of the material classes into which they are classified. The main difference of RBCs compared to dentin is a comparable hardness but a much lower indentation modulus, emphasizing further development potential.

## 1. Introduction

The constant expansion of scientific knowledge, underpinned by the further development of existing filling materials and the introduction of completely new ones, leads to changes in dental treatment concepts. In the chronological development of resin-based composites (RBCs), bulk-fill RBCs represent the most recent development. They were developed with the aim of significantly reducing the treatment time by no longer having to build up RBC fillings in 2 mm layers, but instead applying them in 4 to 5 mm thick layers. In addition, the risk of introducing defects such as voids or contaminants between two-layered increments is also reduced. Since the classic incremental-layering technique is time-consuming and technique-sensitive, especially in large-volume posterior cavities, the bulk-fill application has attracted interest as an alternative. Today the concept is established, and a clinical time saving of approx. 20% compared to the conventional, incremental restoration technique is documented [1].

This alternative bulk-filling concept originally arose from the idea of combining bulk-fill RBCs in two different viscosities. Although the first light-curing, high-viscosity bulk-fill RBC (QuiXfil, Dentsply) [2] came onto the market 20 years ago, the bulk-fill concept only gained clinical acceptance years later with the marketing of the low-viscosity (flowable) bulk-fill RBCs. The concept of a bulk-fill flowable RBC was inspired by the habit of many dentists of using more user-friendly and adaptable conventional flowable RBCs, when the clinical situation allows. A flowable material requires fewer fillers, making it more translucent, thus improving depth of cure by allowing light to penetrate deeper layers, a fact closer to the bulk-filling concept. On the other hand, a lower amount of filler also lowers the mechanical properties of the material [3], which limits the clinical indication. Therefore, when used in deep and large cavities, flowable bulk-fill RBCs require capping with a higher-filled material. Bulk-fill materials with higher viscosity, which can also be applied without capping, followed the path thus opened for the bulk-fill technique. In addition to a still-reduced amount of filler compared to regular RBCs [4], a larger filler size [5], and a reduction in color pigments also helped increase the depth of cure by increasing translucency, as no other changes were made in the chemical composition of the first bulk-fill compared to the conventional RBCs [6]. However, it was quickly recognized that the increased translucency could potentially lead to less esthetic integration of the restorations. Occasionally, a certain gray cast of the restoration was observed as well as a low masking potential of dark tooth discolorations. Although this may not be a problem in the posterior region, it must be taken into account in individual cases, e.g., on the mesial surfaces of premolars.

The described aesthetic deficit has been addressed in following materials either by adding suitable opacifiers (nanoparticles of zirconium oxide or ytterbium trifluoride) or by focusing the polymerization process on mechanisms that are less dependent on reduced translucency. These include the addition of new, more efficient photo-initiators (germanium-based) [7] to supplement traditional ones (such as camphorquinone (CQ)/amine and acyl-phosphine oxide) or the modification of the polymerization mechanism [8]. The germanium based photoinitiators are Norrish Type I sensitizers that do not require a co-initiator as, in contrast to the H-abstraction and electron transfer mechanism in CQ-amine systems (Norrish Type II sensitizer), they are able to generate two radicals by α-cleavage upon exposure to light [7]. This makes them more efficient compared to CQ as both radicals can initiate radical polymerization compared to only one in CQ [7]. Their reactivity is also classified as higher compared to CQ, which is due to the higher molar extinction coefficient (ε_λ_) and relates to a high probability of light absorption at a given wavelength. Germanium-based initiators can be partially activated by the blue light, but they need shorter wavelength light (violet) to reach their full potential [7]. This aspect requires more caution in choosing the light curing device, which should be a violet-blue LED or a halogen-light curing device [9].

Another advancement, which also relies less on reduced translucency, includes a modification of the polymerization mechanism, namely reversible addition-fragmentation chain transfer (RAFT) polymerization [8,10]. It represents the most recent of the living/controlled free-radical methodologies [11] and offers the advantage that the chemical composition of methacrylate-based RBCs can be maintained while essentially adding a suitable RAFT agent [12] to a conventional free-radical polymerization.

Bulk-fill RBCs with a different, optically perceptible translucency than the tooth in the uncured state were also developed, to offer an additional clinical simplification of the application. This allows for a clearer distinction between the RBC and the tooth when applied in the cavity, while the translucency of the RBC decreases after curing [13] and is no longer visually different. Recently introduced bulk-fill RBCs further simplify clinical application by combining bulk-fill placement with universal chromatic concepts [14], eliminating the time spent searching for a suitable shade to aesthetically match a clinical situation.

Additional features to support simplified restoration concepts, such as bioactive properties [15,16,17], have also been implemented in bulk-fill RBCs, while the fact that some materials have the option of dual-curing no longer limits bulk-fill placements [16,17] to 4–5 mm, as is required in light-cured bulk-fill RBCs [18,19]. As a recent development, these materials have been little studied and not directly compared to materials with the same clinical indication.

Clinical evaluation of bulk-fill RBC restorations to date shows statistically similar failure rates to conventional restorations that have been filled incrementally (2 mm increments) [20]. While data on recently introduced materials are not yet available, many bulk-fill RBCs have been extensively studied in vitro. Several hundred studies published to date document product-specific physical [4] and mechanical properties [6]. The great variability of the properties nevertheless reflects that of conventional RBCs [6]. The same applies to the quality of the bond to the tooth structure [21,22] or biocompatibility [23,24]. In vitro studies also confirm that an increment thickness of 4 mm should not be exceeded for most light-cured bulk-fill RBCs [18,19]. One of the postulated advantages of a bulk-fill placement, the reduction of polymerization shrinkage stress compared to incremental application, could not be clearly proven [25,26,27].

The present study aims to offer a direct comparison of bulk-fill RBCs belonging to the different development and clinical application concepts described above, and to relate them to the properties of dentin, the tooth structure they are intended to replace. This direct comparison is possible through instrumented indentation techniques.

It is therefore hypothesized that bulk-fill RBCs will perform similarly to dentin in terms of elastic–plastic mechanical behavior when measured under similar conditions and within the material category for which they are marked.

## 2. Materials and Methods

### 2.1. Materials

A total of 18 commercially available bulk-fill resin-based composites (bulk-fill RBCs) were characterized in terms of their micromechanical properties and related to properties measured under identical conditions in human dentin (Table 1). The materials are divided into the categories of low-viscosity light-curing bulk-fill RBCs (Table 1a), high-viscosity light-curing bulk-fill RBCs (Table 1b) and dual-curing bulk-fill RBCs (Table 1c).

### 2.2. Methods

#### 2.2.1. Specimen Preparation

A total of 108 samples (2 mm × 2 mm × 18 mm) were prepared, corresponding to six samples for each of the 18 bulk-fill RBCs analyzed, following the recommendation of ISO 4049:2019 [28]. The unpolymerized material was therefore filled into a white polyoxymethylene mold and pressed between two glass plates that were separated from the material by transparent polyacetate films. The light-curing protocol included irradiation for 20 s on the top and bottom of the samples according to the standard mentioned above, with three exposures overlapping an irradiated section by no more than 1 mm of the diameter of the light guide to avoid multiple exposures. A blue LED (Light-Emitting Diode) LCU (Light Curing Unit) (Bluephase^®^ Style, Ivoclar Vivadent, Schaan, Liechtenstein) was used for curing. Immediately after demolding, the samples were stored in distilled water at 37 °C for 24 h. After that, the surface that had received the first exposure was wet-ground with silicon carbide paper (grain size p1200, p2500 and p4000, LECO) and polished with a diamond suspension (average grain size: 1 µm) until the surface was shiny (grinding machine EXAKT 400CS Micro Grinding System, EXAKT Technologies Inc., OK, USA). In addition, tooth samples of 2 mm thickness were obtained from 10 caries-free human molars. For this purpose, a slice was cut per tooth perpendicular to the longitudinal axis of the tooth and polished as described above.

#### 2.2.2. Instrumented Indentation Test (IIT): Quasi-Static Approach (ISO 14,577 [29])

The micromechanical properties were evaluated using an automated microindenter (FISCHERSCOPE^®^ HM2000, Helmut Fischer, Sindelfingen, Germany) equipped with a Vickers diamond tip. Six indentations were performed in each bulk-fill RBC and tooth specimen. Measurements were performed under force control by recording the indentation depth and the indentation force during each indentation cycle, consisting of increasing the force from 0.4 mN to 1000 mN within 20 s at constant speed, holding the maximum force for 5 s and finally reducing it to zero within 20 s. Each indentation created an impression, and the projected contact area of the indenter (A_c_) and the surface area of the indentation under the applied test load (A_s_) were determined for further parameter calculations. The projected indenter contact area (A_c_) was determined from the force–indentation depth curve considering the indenter correction according to the model of Oliver and Pharr and described in ISO 14,577 [29]. Therefore, the indenter area function was calibrated to two different materials (sapphire and fused silica) with consistent and known material properties. Corrections obtained from the tip calibration were then used for further computational data analysis. The elastic and plastic deformation was described by the universal hardness (also known as Martens hardness = F/A_s_(h)) and was calculated by dividing the test load by the surface area of the indentation under the applied test load (As). As a measure of the resistance to plastic deformation, the indentation hardness was calculated from the maximum indentation force and the projected indenter contact area (H_IT_ = F_max_/A_c_). This parameter was then converted to the more familiar Vickers hardness (HV = 0.0945 × H_IT_). Finally, the indentation modulus (E_IT_) was calculated from the slope of the tangent of the indentation–depth curve at maximum force.

### 2.3. Statistical Analysis

All variables were normally distributed, allowing a parametric approach to be used. A multifactor analysis of variance was applied to compare the parameters of interest (Martens and Vickers hardness, and indentation modulus). Results were compared using one-way and multiple-way analysis of variance (ANOVA) and Tukey honestly significant difference (HSD) *post hoc*-test using an alpha risk set at 5%. A multivariate analysis (general linear model) evaluated the influence of the parameters *filler volume, filler weight* and *material category* on the analyzed properties. The partial eta-squared statistic reported the practical significance of each term, based on the ratio of the variation attributed to the effect. Larger values of partial eta-squared (η_P_^2^) indicate a greater amount of variation accounted for by the model (SPSS Inc. Version 27.0, Chicago, IL, USA).

## 3. Results

A multifactorial analysis indicates a significant (*p* < 0.001) and very strong effect of the parameters *filler volume* and *filler weight* on the measured properties, while the effect of *material category* was consistently lower. The *filler volume* had the greatest impact on indentation modulus E_IT_ (η_P_^2^ = 0.918), closely followed by Martens HM (η_P_^2^ = 0.900) and Vickers hardness HV (η_P_^2^ = 0.851). A similar ranking of the parameters was also observed with regards to the effect strength of the filler weight (E_IT_: η_P_^2^ = 0.894; HM: η_P_^2^ = 0.878 and HV: η_P_^2^ = 0.832). For material category, e.g., low- and high-viscosity light-cured and dual-cured bulk-fill RBCs, the ranking of the measured parameters with regard to the effect strength exerted remained the same, but the effect was considerably lower (E_IT_: η_P_^2^ = 0.578; HM: η_P_^2^ = 0.566 and HV: η_P_^2^ = 0.540).

The lowest filler weight and volume were found in the light-cured low-viscosity bulk-fill RBC category (mean value: 67.7% and 45.4%, respectively) with a range of 64.5% (Filtek™ Bulk Fill Flowable Restorative and Filtek Bulk Fill) to 75% (x-tra base) for the filler weight and 38% (Venus Bulk Fill) to 61% (x-tra base) for the filler volume. For bulk-filled high-viscosity light-cured RBCs, filler weight ranged from 76.5 (Filtek One and SonicFill 2) to 87% (Beautifil Bulk restorative), while the filler volume ranged from 58.5 (Filtek One and SonicFill 2) to 74.5% (Beautifil Bulk restorative).

Within a material category, a one-way ANOVA showed that only the material x-tra base achieved a significantly higher HM value compared to dentin in the low-viscosity light-cured bulk-fill RBC category (Figure 1a), while all bulk-fill high-viscosity RBCs achieved a significantly higher HM value (Figure 2a). In dual-cured bulk-fill RBCs, Cention N behaved similarly to dentin, and both showed significantly lower HM compared to Cention Forte (Figure 1c).

The statistical analysis is somewhat similar for the Vickers hardness, while dentin has similar HV values as Filtek Supreme XT flow and Tetric EvoFlow bulk-fill in the light-cured low-viscosity bulk-fill RBCs (Figure 2a) and lower values than all analyzed high-viscosity bulk-fill RBCs (Figure 2b). For dual-cured bulk-fill RBCs, the ranking was similar to that for HM (Figure 2c).

A completely different situation was observed for the indentation modulus. Dentin ranked highest among all light-cured low-viscosity bulk-fill RBCs (Figure 3a) and was surpassed only by X-tra Fil and QuixFil among the high-viscosity bulk-fill RBCs (Figure 3b). Cention Forte still outperformed Cention N in dual-cured bulk-fill RBCs, however, both had a lower indentation modulus compared to dentin (Figure 3c).

## 4. Discussion

The health challenge posed by tooth decay places an enormous burden on healthcare systems around the world [30]. Research is therefore focused on cost-effective materials and/or cost-effective restorative techniques to make dental treatment affordable for everyone. In this line, adhesive, tooth-colored restorations, with which lost tooth tissue and caries defects can be replaced quickly and in just one step e.g., in bulk, are enjoying increasing popularity.

Since dental materials are intended to replace the tooth structure, their properties should be designed accordingly. However, test methods that allow a direct comparison of both substrates are limited due to the small size and the heterogeneity of the tooth samples. In the present study, we opted for the quasi-static micro-indentation method, which, with indentation depths of 4–10 µm and standardized methods, not only enables a direct material and tooth substrate comparison, but also comparison with data collected over time in large databases that contain many clinically successful materials [3]. The latter is particularly important for materials that have not yet been analyzed in clinical studies. In addition to the plastic deformation defined by the Vickers hardness, the depth-sensing indentation method used also allows the elastic deformation contained in the universal hardness parameter to be assessed. Additionally, the tested indentation modulus is of high relevance as it has previously been found to correlate with the modulus of elasticity measured in three-point bending test [31], which is considered one of the most important test methods for dental materials and one of the few that records a correlation with the clinical behaviour of the materials [32]. Even if measured data have not yet been correlated with the clinical performance of the materials, since many materials have not yet been evaluated in clinical studies, they are valuable input parameters for further FEM analysis, which allow us to predict the effect of the material within a restoration of a given size and geometry, and the development of appropriate restoration techniques.

The classification of RBCs into material categories is rightly controversial and is very often based on advertising strategies. While the low-viscosity bulk-fill RBCs are generally characterized by lower filler content compared to the high-viscosity bulk-fill category, individual materials may not reflect their classification system (Table 1). This statement is again confirmed by the results of this study, since the effect of the material category on the measured properties is significantly lower compared to the effect of the filler content, confirming previous studies [3].

The dentin substrate tested in the present study should not only allow a comparison with the analyzed materials, but also a ranking of all analyzed materials, since it serves as a reference in all graphic representations (Figure 1, Figure 2 and Figure 3). Generally speaking, dentin is a porous, mineralized connective tissue with an organic matrix (collagenous proteins), an inorganic component (hydroxyapatite), and a microstructure composed of different types of dentin with its own peculiarities [33]. In the case of the Vickers indentation, the indentation size (indentation diagonal) can be rated at around seven times the indentation depth measured at maximum load. With indentation depths of 9 µm to 10 µm and corresponding indentation sizes of 63 µm to 70 µm, the properties measured in dentin represent the properties of the substrate at the measurement location and not the properties of the individual components (collagen, hydroxyapatite). Since the hardness and modulus of elasticity depend on the measuring location and age and are higher at the dentin–enamel junction and lower towards the pulp [34], indentations were performed arbitrarily in different teeth and regions in order to do justice to this diversity. This allows value ranges to be specified for the properties of the dentin and thus captures the variation of the entire substrate area. Note that there is a discrepancy in the ranking of materials and dentin as a function of the property analyzed, highlighting important differences in the mechanical behavior of the restorative material and the tooth substrate. In terms of hardness, the dentin ranks somewhat in the middle of the analyzed materials, showing statistically similar hardness compared to three other materials, lower values compared to nine and higher values compared to six. It is worth noting that the three materials statistically showing the same hardness values as dentin belong to all three categories examined, further confirming the previous statement about the meaningless categorization of materials. In comparison, the indentation modulus of dentin was higher than that of 16 of the 18 materials studied. Ideally, the elastic modulus of a restorative material should match the biologically replaced tissue to reduce the risk of creating a stress shielding effect during chewing. While an excellent correlation between the amount of inorganic filler and the indentation modulus was found for RBCs [3], the discrepancy with dentin cannot be explained by the hydroxyapatite content (70% by weight, 40–45% by volume) [33], which is lower as the inorganic filler amount in many of the analyzed materials and rather comparable to the flowable analyzed materials (Table 1), but with the microstructure. When considering the quotient of inorganic weight percent to volume percent in dentin vs. RBCs tested (Table 1), it is important to note that the density of dentin is apparently higher than that of the materials tested, confirming the influence of the microstructure on the measured properties.

The analyzed light-cured low-viscosity bulk-fill RBCs reflect a time scale in the development of the materials. The initial development of such materials, starting chronologically with SureFil^®^ SDR™ flow, aimed to create a material with excellent rheological properties to allow flow in narrow and deep cavities where condensing a material is not possible, and/or to serve as a flowable lining. The claimed that it reduced the polymerization shrinkage required for proper adaptation to the cavity after curing was confirmed as it was found that the shrinkage stress after polymerization not only compared to regular flowable materials but also compared to higher-filled RBCs, or the low shrinkage silorane was lower [35]. This effect is related to the incorporation of a polymerization modulator in a high-molecular-weight urethane-based methacrylate resin capable of retarding gelation and thus reducing polymerization shrinkage without impairing the degree of conversion [35]. With regard to the analyzed micromechanical properties, it ranks in the middle of the materials developed later, although it is true for all materials in this category that the lower indentation modulus compared to dentin does not recommend them for use in larger cavities without capping.

Another distinctive feature of many bulk-fill RBCs—e.g., SDR, Venus Bulk Fill, Venus Bulk One, Beautifil Bulk Flow—is that the base monomer bisphenol A-glycidyl methacrylate (Bis-GMA) has been completely replaced by less-viscous dimethacrylates such as urethane dimethacrylate (UDMA) and its derivatives, triethylene glycol dimethacrylate (TEGDMA) or ethoxylated bisphenol-A-dimethacrylate (EBPDMA), which can form more flexible polymers than Bis-GMA [36,37].

The progressive developments observed in regular RBCs are increasingly reflected in bulk-fill materials as well. Here, too, there is a trend of modifying RBCs to meet the increased demand for bioactive materials that can prevent the recurrence of carious lesions. This was conducted in Beautifil Bulk Flow and Beautifil Bulk Restorative [15] by adding a special filler, a pre-reacted glass-ionomer (PRG), developed and originally described by Roberts et al. [38], that enabled the release of fluoride and other ions such as Na^+^, Sr^2+^, Al^3+^, BO_3_^3−^, and SiO_3_^3−^. Silicate and fluoride are known to act as strong inducers of the remineralization of the dentin matrix [39]. Strontium and fluoride, on the other hand, also improve the acid resistance of teeth by acting on hydroxyapatite to transform it into strontium and fluoroapatite [40]. The difference in the amount of filler between the low- and high-viscosity versions is clearly reflected in all measured parameters.

The ability to leach out caries-protective ions such as phosphate, fluorite, and especially alkali and alkaline earth ions such as calcium [41,42], was also adopted in the development of Cention N and its further development Cention Forte. This ability is caused by the incorporation of alkaline glass fillers (SiO_2_-CaO-CaF_2_-Na_2_O glass) capable of raising the pH and forming fluoroapatite in phosphate-containing media, which explains the low incidence of secondary caries in Cention N [43]. In addition to the bulk-fill application, the strategy of self-adhesiveness was pursued. While Cention N was originally designed to be applied without additional surface pre-treatment, Cention Forte is used with a primer. However, it has been shown that the additional use of an adhesion promoter in Cention N, for example, a universal adhesive, improves the bond strength to dentin [44]. An additional advantage of both materials is that they were designed as dual-cured materials, while the bulk application requirements for both materials proved to be insensitive regardless of whether the material receives additional light or not [16,17]. The additional exposure to light in Cention N was only apparent in the first few minutes of polymerization, in that it was able to accelerate polymerization kinetics and thus shorten the restoration process by curing the material when required, but it did not affect the final mechanical properties or degree of cure [16]. The bioactive potential of Cention N has also been demonstrated [43,45,46] along with its ability to inhibit dentin and enamel demineralization at restoration margins [46]. The mechanical properties of both materials are comparable to those of regular nano- and micro-hybrid RBCs as well as bulk-fill RBCs [16,17]. Direct comparison of both materials in the present study shows superior micromechanical properties in Cention Forte, while the relationship to dentin indicates superior hardness but lower indentation modulus, confirming the behavior observed in most RBCs.

The most recently launched bulk-fill RBC is Venus Bulk Flow One, a material that meets two criteria at once to speed up the restoration process as it is a universal chromatic material that can be applied in bulk [14]. Despite being a less-filled bulk-fill RBC, it is said to have a modified rheology that allows its use in posterior occlusal cavities without the need for covering with a higher-filled RBC. The material was intensively analyzed as an experimental material in its last formulation before being launched [14] and has been proven to meet the requirements of ISO 4049:2019 [28] for materials intended for used in the occlusal areas, as it has a flexural strength > 80 MPa not only within the testing criteria of the standard, but also after aging. However, due to the low modulus of elasticity compared to clinically successful, high-viscosity RBCs, it was advisable to limit the indication of the material to smaller cavities [14]. This is consistent with the present study, as the measured micromechanical properties clearly place the material in the low-viscosity, bulk-fill RBC category.

Since materials have individual properties that differ not only between material categories but also within the material categories to which they belong and the tooth substance, all null hypotheses are rejected.

## 5. Conclusions

Bulk-fill RBCs should not necessarily be classified into material categories but should be considered as materials to be evaluated individually as the properties measured relate to the amount of inorganic filler and not to the material category to which they belong. The ratio of the tested materials to dentin is also individual, with the general conclusion that the hardness of the tested RBCs is somewhat in the range of dentin, but dentin is characterized by an indentation modulus superior to most of them. This aspect can be useful in further material development, for a better adaptation of the material properties to those of the dentin.

## Figures and Tables

**Figure 1 materials-15-08095-f001:**
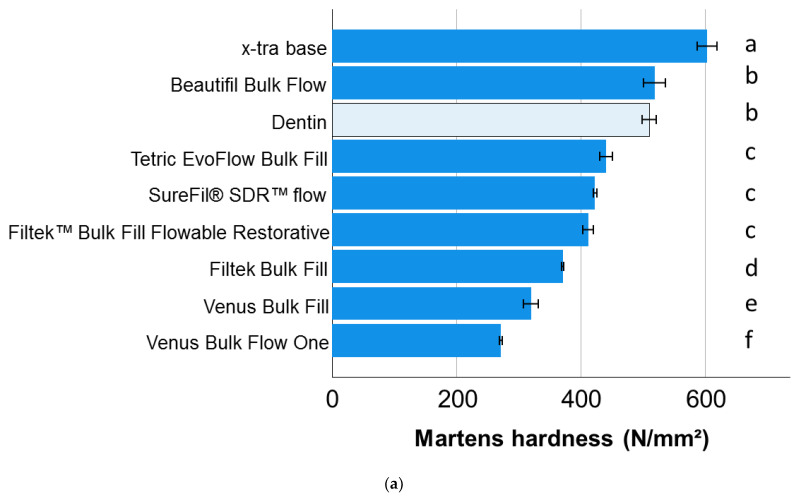
Martens hardness (means with 95% confidence interval) as a function of RBC and bulk-fill category: (**a**) light-cured, low-viscosity bulk-fill RBCs; (**b**) light-cured, high-viscosity bulk-fill RBCs; and (**c**) dual-cured bulk-fill RBCs. Letters indicate homogeneous groups within each material category; Tukey’s post-hoc test (α = 0.05).

**Figure 2 materials-15-08095-f002:**
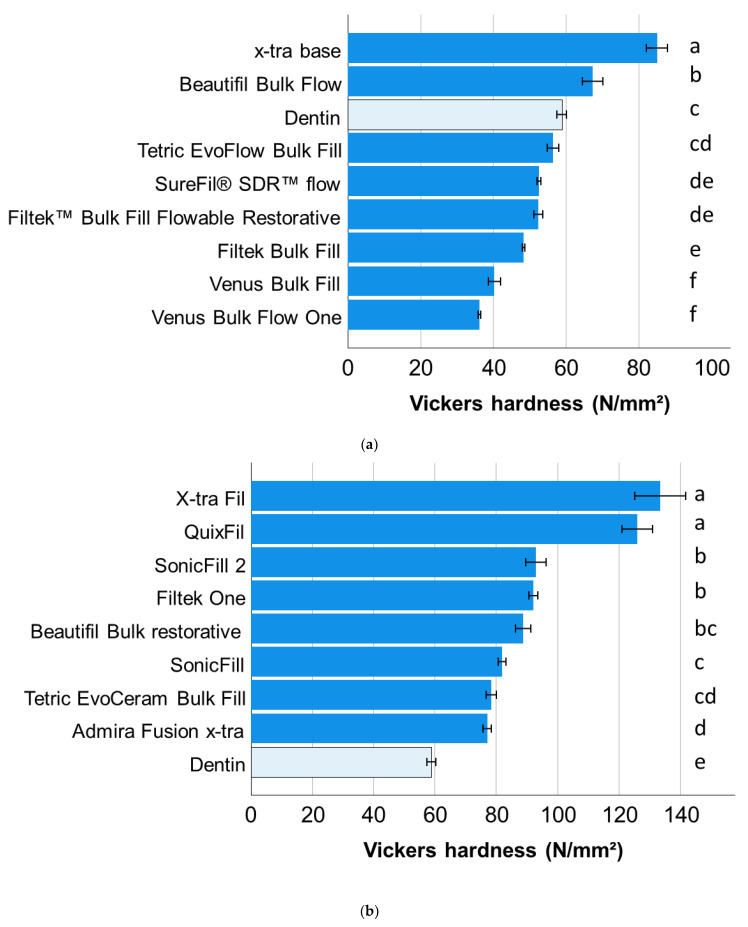
Vickers hardness (means with 95% confidence interval) as a function of RBC and bulk-fill category: (**a**) light-cured, low-viscosity bulk-fill RBCs; (**b**) light-cured, high-viscosity bulk-fill RBCs; and (**c**) dual-cured bulk-fill RBCs. Letters indicate homogeneous groups within each material category; Tukey’s post-hoc test (α = 0.05).

**Figure 3 materials-15-08095-f003:**
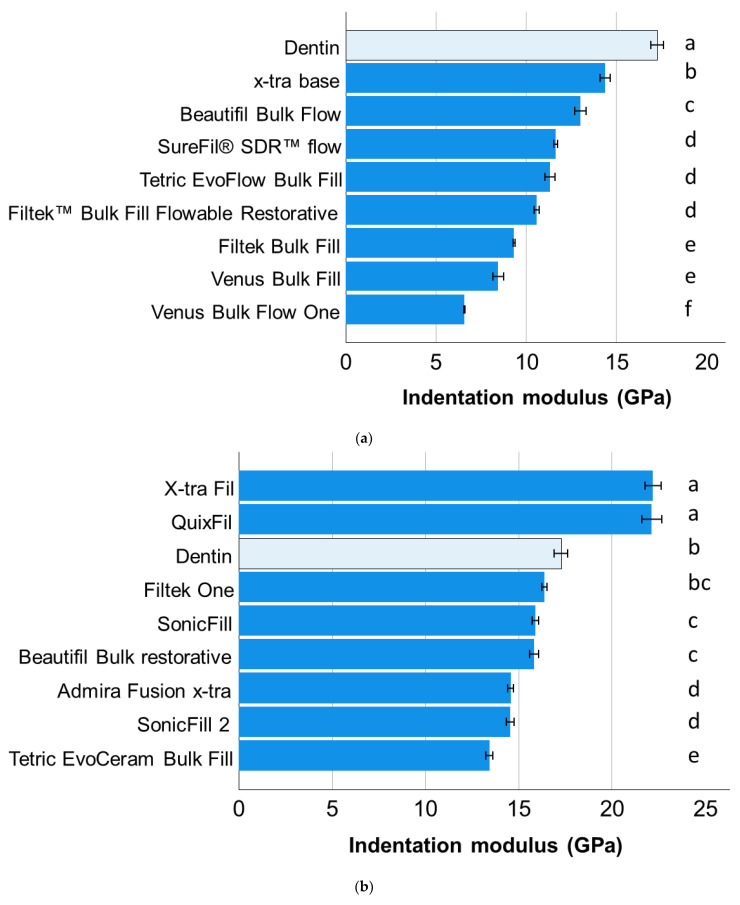
Indentation modulus (means with 95% confidence interval) as a function of RBC and bulk-fill category: (**a**) light-cured, low-viscosity bulk-fill RBCs; (**b**) light-cured, high-viscosity bulk-fill RBCs; and (**c**) dual-cured bulk-fill RBCs. Letters indicate homogeneous groups within each material category; Tukey’s post-hoc test (α = 0.05).

**Table 1 materials-15-08095-t001:** Analyzed bulk-fill RBCs and amount of filler in volume (vol) and weight percent (wt) as reported by the manufacturer. (a) light-cured low-viscosity bulk-fill RBCs; (b) light-cured high-viscosity bulk-fill RBCs; (c) dual-curing bulk-fill RBCs.

RBC	Manufacturer	Lot	Filler wt/vol %
(a)			
Beautifil Bulk Flow	Shofu	121301	72.5/51
Filtek Bulk Fill	3M	N387662	64.5/42.5
Filtek™ Bulk Fill Flowable Restorative	Ivoclar/Vivadent	N692537	64.5/42.5
SureFil^®^ SDR™ flow	Dentsply	100507	68/44
Tetric EvoFlow Bulk Fill	Ivoclar/Vivadent	U12113	68.2/44.4
Venus Bulk Fill	Kulzer	010108	65/38
Venus Bulk Flow One	Kulzer	M010021	65/41
x-tra base	Voco	V 45226	75/61
(b)			
Admira Fusion x-tra	Voco	1527519	84/-
Beautifil Bulk restorative	Shofu	011402	87.0/74.5
Filtek One	3M	N782223	76.5/58.5
QuixFil	Dentsply	100774	85.5/66.4
SonicFill	Kerr	4426994	83.5/-
SonicFill 2	Kerr	5767358	76.5/58.5
Tetric EvoCeram Bulk Fill	Ivoclar/Vivadent	P48872	79–81/60–61
X-tra Fil	Voco	1202359	86/70.1
(c)			
Cention N	Ivoclar/Vivadent	U19921	78.4/
Cention Forte	Ivoclar/Vivadent	ZL08SZ	-/58–59

## Data Availability

Data available on demand.

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
