# Peer review of "Resin-Based Bulk-Fill Composites: Tried and Tested, New Trends, and Evaluation Compared to Human Dentin"

_materials, 2022, doi:10.3390/ma15228095_

Round 1
Reviewer 1 Report
Compact, well written research report .
Author Response
The author would like to thank the reviewer for taking the time to read and critically appraise the manuscript and for their positive, constructive comments on improving the work.
All comments to the corresponding author have been addressed independently below. The authors’ rebuttal is BLUE, and changes to the revised manuscript in light of the Editors’ comments are presented in RED.
Comments and Suggestions for Authors
Compact, well written research report.
Author’s response: We would like to thank the reviewer for the appreciation and the positive evaluation.
Reviewer 2 Report
This paper compared the micro-mechanical properties of bulk-fill Resin-Based Composites from various development and clinical application concepts, as well as the properties of dentin.
The authors' efforts are commendable.
The findings of this article assist clinicians in selecting the best resin-based composite material for the demands of their patients.
The degree of polymerization, on the other hand, is an important factor in determining various mechanical properties of composite materials, and its effect was not determined in this manuscript to support the results.
The current study's findings should be compared to those of previous similar studies.
Author Response
The author would like to thank the Reviewers for taking the time to read and critically appraise the manuscript and for their positive, constructive comments on improving the work.
All comments to the corresponding author have been addressed independently below. The authors’ rebuttal is BLUE, and changes to the revised manuscript in light of the Editors’ comments are presented in RED.
Comments and Suggestions for Authors
This paper compared the micro-mechanical properties of bulk-fill Resin-Based Composites from various development and clinical application concepts, as well as the properties of dentin.
The authors' efforts are commendable.
The findings of this article assist clinicians in selecting the best resin-based composite material for the demands of their patients.
Author’s response: We would like to thank the reviewer for the appreciation and the positive evaluation
The degree of polymerization, on the other hand, is an important factor in determining various mechanical properties of composite materials, and its effect was not determined in this manuscript to support the results.
Author’s response: I strongly agree with this comment. However, the degree of cure can only be compared within a system and is meaningless when comparing different materials. What I mean is that it makes sense to compare the DC within a material at different depths to assess the degree of polymerization. In material comparison, the value of DC is of no relevance because for instance, Ormocers, due to the low cross-ling and large monomers show a DC of max. 30-32%. This, however, does not mean that the material is less cured or the mechanical properties are low. The correlation of DC and mechanical properties requires another study design. To be considered is also the fact that in the present study the materials received the amount of light they needed for adequate polymerization and the aim was to compare the materials among themselves and within their material category and dentin while being adequately cured.
The current study's findings should be compared to those of previous similar studies.
Author’s response: I extended the discussion part with references and comments on this statement, as suggested. E. g: “In the present study, we opted for the quasi-static micro-indentation method, which, with indentation depths of 4-10 µm and standardized methods, not only enables a direct material and tooth substrate comparison, but also a comparison with data collected over time in large databases that contain many clinically successful materials [3]. “
“Several hundred studies published to date document product-specific physical [4] and mechanical properties [6]. The great variability of the properties nevertheless reflects that of conventional RBCs [6].”
Reviewer 3 Report
With the development of bulk fill application, this paper mainly tested the hardness (Vickers and Martens hardness) and indentation modulus (indentation modulus) of resin-based composites (RBC) and materials with less recent research using this technology and compared them with dentin. It was concluded that the hardness of the tested RBC was within the range of dentin, but the indentation modulus of dentin was superior to most materials. The instrumented indentation test in this paper provides relevant information for clinical use.
However, before being accepted for publication, major revision is still required for improvements. My suggestions are listed below:
1. Page 1 – the introduction section
It is suggested that the author increase the logic of the introduction part, distinguish the primary and secondary parts, strengthen the connection with the research content, and finally clarify the research purpose and significance.
2. Page 1, Line 27 – the introduction section
“Since the classic incremental layering technique of resin-based composites (RBCs) is a very time-consuming and technique-sensitive procedure, especially in large-volume posterior cavities, bulk-fill application, e.g. in thick layers of 4-5 mm, has been received with interest as an alternative”
A brief explanation of the process of incremental layering technology here is suggested here so that it can more effectively prove that bulk fill can save time.
3. Page 2, Line 63 – the introduction section
“Another advantage is the fact that they 63 do not require a co-initiator (no amines).”
It is suggested that the author elaborate here on what specific effects the germanium-based photoinitiator without amines has on repairing the aesthetic defect.
4. Page 2, Line 46 to 57 – the introduction section
This part introduces the influence and improvement of material transparency on the restoration, which is less related to the hardness and indentation modulus of RBC tested below.
5. Page 2, Line 82 – the introduction section
“Additional features to support simplified restoration concepts, such as bioactive properties, have also been implemented in bulk-fill RBCs, while the fact that some materials have the option of dual-curing no longer limits bulk-fill placements to 4- 5 mm like it is required in light-cured bulk-fill RBCs”
I believe that it would be better for the author to clarify the purpose and significance of the comparison with this material.
6. Page 10, Line 247 –the discussion section
“Moreover, the depth sensing indentation method used allow to assess in addition to the plastic deformation defined by the Vickers hardness also the elastic deformation included in the universal hardness parameter.”
It is recommended that the authors explain the definitions of Martens hardness, Vickers hardness, and indentation modulus in this section and what specific effects they have on the function of the prosthesis in detail.
7. Page 10, Line 279 – the discussion section
“Generally speaking, dentin is a porous, mineralized connective tissue with an organic matrix (collagenous proteins) and an inorganic component (hydroxyapatite) and a microstructure composed of different types of dentin with its own peculiarities.”
The author mentioned the microstructure of dentin here. I think it would be better to introduce the microstructure first and then explain the influence of the microstructure on the indentation modulus, so that it can be more logically improved.
8. Page 10, Line 280 – the discussion section
“When considering the quotient of inorganic weight percent to volume percent in dentins. RBCs tested (Table 1), it is important to note that the density of dentin is apparently higher than that of the materials tested. In dentin, indentations were therefore carried out arbitrarily in different teeth and regions in order to do justice to this diversity.”
It is suggested that the author clarify the relationship or influence of dentin density higher than that of test materials on indentations in different areas of teeth
9. Page 11, Line 306 – the discussion section
“Here, too, there is a trend to modify RBCs to meet the increased demand for bioactive materials that can prevent the recurrence of carious lesions.”
I think it would be better for the author to explain the principle or mechanism of how the resin has the biological activity to prevent the recurrence of carious lesions in detail.
10. The conclusion is that dentin’s indentation modulus is better than most RBCs. What are the consequences if RBC is used? What better suggestions does the author have for the clinical transformation of results or material improvement? It is suggested to join the author's future outlook in the previous section, such as discussion.
11. The language and words used in this manuscript are suggested for improvement with more precise and clearer expressions.
Author Response
The author would like to thank the Reviewers for taking the time to read and critically appraise the manuscript and for their positive, constructive comments on improving the work.
All comments to the corresponding author have been addressed independently below. The authors’ rebuttal is BLUE, and changes to the revised manuscript in light of the Editors’ comments are presented in RED.
Comments and Suggestions for Authors
With the development of bulk fill application, this paper mainly tested the hardness (Vickers and Martens hardness) and indentation modulus (indentation modulus) of resin-based composites (RBC) and materials with less recent research using this technology and compared them with dentin. It was concluded that the hardness of the tested RBC was within the range of dentin, but the indentation modulus of dentin was superior to most materials. The instrumented indentation test in this paper provides relevant information for clinical use.
Author’s response: We would like to thank the reviewer for the appreciation and the positive evaluation.
However, before being accepted for publication, major revision is still required for improvements. My suggestions are listed below:
- Page 1 – the introduction section
It is suggested that the author increase the logic of the introduction part, distinguish the primary and secondary parts, strengthen the connection with the research content, and finally clarify the research purpose and significance.
Author’s response: I have restructured large parts of the introduction. The introduction presents the technological achievements in the analyzed material category, as materials from all the aspects described have been incorporated into the study design. It first describes the benefits of using bulk filling instead of incremental placement. Then it focuses on the chronological developments: highly translucent materials to allow light to reach deeper layers; then the recognition of the aesthetic loss due to high translucency and improvement of this through the use of new initiators and the modification of the polymerization mechanism (RAFT). Finally, the latest development was briefly described, which includes the new trends in the development of RBCs, namely to make them bioactive. It is a narrative of the main developments, so there are no primary and secondary parts. Hope the restructuring and additions have made the text clearer.
- Page 1, Line 27 – the introduction section
“Since the classic incremental layering technique of resin-based composites (RBCs) is a very time-consuming and technique-sensitive procedure, especially in large-volume posterior cavities, bulk-fill application, e.g. in thick layers of 4-5 mm, has been received with interest as an alternative”
A brief explanation of the process of incremental layering technology here is suggested here so that it can more effectively prove that bulk fill can save time.
Author’s response: Changes have been made accordingly. Please also note the references provided which certify that the bulk fill application technique saves time compared to the regular application technique of RBCs. E.g. ” Vianna-de-Pinho, M.G.; Rego, G.F.; Vidal, M.L.; Alonso, R.C.B.; Schneider, L.F.J.; Cavalcante, L.M. Clinical Time Required and Internal Adaptation in Cavities restored with Bulk-fill Composites. J Contemp Dent Pract 2017, 18, 1107-1111”.
- Page 2, Line 63 – the introduction section
“Another advantage is the fact that they 63 do not require a co-initiator (no amines).”
It is suggested that the author elaborate here on what specific effects the germanium-based photoinitiator without amines has on repairing the aesthetic defect.
Author’s response: I extended the explanation, as suggested. However, the nature of the initiator is not intended to repair the aesthetic defect. Being more efficient compared to CQ, it requires less light to activate and as a result, the material does not need to be as translucent. This is the indirect effect on the aesthetics and not the initiator per se. I have deepened this aspect, please consider market changes in the manuscript.
- Page 2, Line 46 to 57 – the introduction section
This part introduces the influence and improvement of material transparency on the restoration, which is less related to the hardness and indentation modulus of RBC tested below.
Author’s response: Not at all! Any modification in chemical composition and filler amount affects the mechanical properties. As explained, the translucency is induced by using less filler - this not only has a direct effect but also has an essential effect on hardness and indentation modulus. Please consider the results of the statistical analysis that indicate a great correlation between the filler amount and measure properties.
- Page 2, Line 82 – the introduction section
“Additional features to support simplified restoration concepts, such as bioactive properties, have also been implemented in bulk-fill RBCs, while the fact that some materials have the option of dual-curing no longer limits bulk-fill placements to 4- 5 mm like it is required in light-cured bulk-fill RBCs”
I believe that it would be better for the author to clarify the purpose and significance of the comparison with this material.
Author’s response: As explained in the first comment, the analyzed materials belong to all these developments. For the first time, the study enables a comparison between the various material developments, and in addition, it relates them to dentin. This was the aim of the presented study. I restructured this paragraph for more clarity.
- Page 10, Line 247 –the discussion section
“Moreover, the depth sensing indentation method used allow to assess in addition to the plastic deformation defined by the Vickers hardness also the elastic deformation included in the universal hardness parameter.”
It is recommended that the authors explain the definitions of Martens hardness, Vickers hardness, and indentation modulus in this section and what specific effects they have on the function of the prosthesis in detail.
Author’s response: The parameters Martens hardness, Vickers hardness, and indentation modulus have been explained in the section material and methods: “The elastic and plastic deformation was described by the universal hardness (also known as Martens hardness = F/As(h)) and was calculated by dividing the test load by the surface area of ​​the indentation under the applied test load (As). As a measure of the resistance to plastic deformation, the indentation hardness was calculated from the maximum indentation force and the projected indenter contact area (HIT = Fmax/Ac). This parameter was then converted to the more familiar Vickers hardness (HV = 0.0945 x HIT). Finally, the indentation modulus (EIT) was calculated from the slope of the tangent of the indentation depth curve at maximum force.
Also, please consider that we are analyzing here restorative filling materials, not prosthesis. I extended the explanation in the discussion and made attentive, to which parameters correlate to clinical behavior and where the limits are.
- Page 10, Line 279 – the discussion section
“Generally speaking, dentin is a porous, mineralized connective tissue with an organic matrix (collagenous proteins) and an inorganic component (hydroxyapatite) and a microstructure composed of different types of dentin with its own peculiarities.”
The author mentioned the microstructure of dentin here. I think it would be better to introduce the microstructure first and then explain the influence of the microstructure on the indentation modulus, so that it can be more logically improved.
Author’s response: I followed your recommendation and also have extended the explanation on how properties are varying in dentin and the reason we collect our data randomly. Please consider marked modifications in the manuscript text.
- Page 10, Line 280 – the discussion section
“When considering the quotient of inorganic weight percent to volume percent in dentins. RBCs tested (Table 1), it is important to note that the density of dentin is apparently higher than that of the materials tested. In dentin, indentations were therefore carried out arbitrarily in different teeth and regions in order to do justice to this diversity.”
It is suggested that the author clarify the relationship or influence of dentin density higher than that of test materials on indentations in different areas of teeth
Author’s response: We have expanded the statement as mentioned above.
- Page 11, Line 306 – the discussion section
“Here, too, there is a trend to modify RBCs to meet the increased demand for bioactive materials that can prevent the recurrence of carious lesions.”
I think it would be better for the author to explain the principle or mechanism of how the resin has the biological activity to prevent the recurrence of carious lesions in detail.
Author’s response: there is not the resin, but the fillers that enable a biological activity to prevent recurrent carious lesions. A brief explanation and references that describe the mechanisms were introduced. However, please consider that these aspects are very complex and divagate from the topic of the paper to offer a mechanical properties comparison.
- The conclusion is that dentin’s indentation modulus is better than most RBCs. What are the consequences if RBC is used? What better suggestions does the author have for the clinical transformation of results or material improvement? It is suggested to join the author's future outlook in the previous section, such as discussion.
Author’s response: This is one of the conclusions, yes, actually the last of them, as the aim was firstly to characterize the materials within the category they are defined, the material categories among them and finally to compare them with the dentin.
Regarding the comparison with the dentin: When designing restorative materials, the focus is placed on some complex, mostly antagonist procedures/parameters (strength, shrinkage, aesthetic, curing depth, degree of cure, biocompatibility, etc). The knowledge gained here, that dentin behaves differently (comparable hardness – higher modulus), should encourage fine-tuning also these aspects when designing new materials, since these differences have so far received little attention and are new aspects. I extended the explanation in the manuscript.
- The language and words used in this manuscript are suggested for improvement with more precise and clearer expressions.
Author’s response: We have worked through the manuscript and improved it in this regard. Please see the highlighted changes in the manuscript.
Round 2
Reviewer 3 Report
I don not have any more questions or/and suggestions,